# User-Centered Redesign of Monitoring Alarms: A Pre–Post Study on Perception, Functionality, and Recognizability Following Real-Life Clinical Implementation

**DOI:** 10.3390/healthcare13233033

**Published:** 2025-11-24

**Authors:** Cynthia Hunn, Christoph B. Nöthiger, Julia Braun, Yoko Sen, Avery Sen, Samira Akbas, Matthias Hoffmann, Elena Neumann, Greta Gasciauskaite, David W. Tscholl, Tadzio R. Roche

**Affiliations:** 1Institute of Anesthesiology and Perioperative Medicine, University Hospital Zurich, 8091 Zurich, Switzerland; cynthiahunn98@gmail.com (C.H.); christoph.noethiger@usz.ch (C.B.N.); samira.akbas@usz.ch (S.A.); matthoffmann@gmx.net (M.H.); elena.neumann@usz.ch (E.N.); greta.gasciauskaite@usz.ch (G.G.); david.tscholl@usz.ch (D.W.T.); 2Departments of Epidemiology and Biostatistics Epidemiology, Biostatistics and Prevention Institute, University of Zurich, 8006 Zurich, Switzerland; julia.braun@uzh.ch; 3Sen Sound, LLC Falls, Church, VA 22046, USA; yoko@sensound.space (Y.S.); avery@sensound.space (A.S.)

**Keywords:** anesthesia, auditory perception, alarm systems, equipment design, monitoring, physiologic, patient safety, sound, ergonomics

## Abstract

**Background:** Auditory alarms in patient monitoring are vital for clinical safety, but their harsh acoustic properties and high frequency contribute to stress, alarm fatigue, and reduced acceptance among healthcare staff. In collaboration with Sen Sound, Philips redesigned its alarm sounds to reduce auditory harshness, particularly for low- and medium-priority alarms, while preserving the salience of high-priority alerts. This study evaluated the impact of these refined alarm sounds in a real-world clinical setting. **Objective:** The goal was to determine whether anesthesia professionals perceive the refined Philips alarm sounds as more pleasant, clinically appropriate, and reliably recognizable compared with the traditional sounds. **Methods:** We conducted a single-center, pre–post intervention study at the University Hospital Zurich, Switzerland. Anesthesia providers assessed traditional and refined Philips alarm sounds with respect to perceived sound appeal, perceived functionality, and recognition accuracy. The primary outcome (sound appeal) was tested for superiority; using mixed-effects regression models. **Results:** Seventy-seven participants completed both study phases. Refined alarm sounds significantly improved perceived sound appeal (mean difference +0.51; 95% CI, 0.37–0.64; *p* < 0.001), while perceived functionality showed a small decrease (mean difference −0.15; 95% CI, −0.27 to −0.03). Recognition accuracy for low- and medium-priority alarms was higher with traditional sounds (low: 95.2% vs. 87.5%, *p* = 0.002; medium: 81.1% vs. 62.0%, *p* < 0.001), while high-priority alarms were more accurately identified with refined sounds (89.0% vs. 81.4%, *p* = 0.002). Overall, 71% of participants preferred the refined sounds, and 92% supported further development. **Conclusions:** Refined alarm sounds reduced perceived harshness and improved auditory comfort for anesthesia providers, but were associated with slightly lower perceived functionality and mixed recognition accuracy. High-priority alarms were identified more reliably, whereas low- and medium-priority alarms were less distinctly recognized, indicating a limited trade-off between sound appeal and clarity that primarily affected lower-priority signals. These findings suggest that while refinement can enhance the auditory environment, further development, potentially incorporating auditory icons or voice-based alerts, will be needed to optimize both user experience and patient safety in clinical practice.

## 1. Introduction

Alarm sounds in patient monitoring are primarily intended to enhance patient safety by alerting clinicians to potentially critical changes in physiological status [1]. To fulfill this function effectively, alarm signals must be clearly distinguishable and capable of eliciting timely and appropriate clinical responses [2,3]. Historically, alarms were intentionally designed to be loud and attention-grabbing; however, this approach often resulted in sounds perceived as harsh and unpleasant [4]. The high frequency of alarms, combined with limited clinical relevance and poor specificity [5,6,7,8], contributes to increased cognitive workload for healthcare providers [9], promotes the development of alarm fatigue [10], and may even pose direct risks to patient safety through desensitization, delayed responses, and increased risk for delirium [11,12,13].

In response to these issues, the International Electrotechnical Commission (IEC) updated the global standard for medical equipment alarms (IEC 60601-1-8) in 2020 [2]. Among the updates was the recommendation to use auditory icons, sounds that carry intuitive meaning [14]. Additionally, scientists have explored further alternative acoustical alarm systems, such as voice alerts and spearcons (speech-based auditory cues) [15,16], as well as methods reduce the overall alarm sound frequency [17,18,19]. Although these approaches have shown promising results in experimental and simulation studies for improving care provider situation awareness [20,21], they have not yet been implemented in clinical patient monitoring practice.

Philips Healthcare (Koninklijke Philips N.V., Amsterdam, The Netherlands) has taken a first step toward addressing the shortcomings of conventional medical alarm sounds by redesigning the auditory alerts of its IntelliVue monitoring systems [22]. This redesign was carried out in collaboration with Sen Sound (Falls Church, VA, USA), a company specializing in user-centered sound design. The aim was to enhance auditory comfort, thereby reducing uncertainty and mitigating unnecessary stress for both clinicians and patients [23]. To achieve this, acoustic features such as pitch, timbre, tone intervals, and amplitude envelopes were systematically modified, particularly for low- and medium-priority alarms, to reduce auditory harshness. In contrast, high-priority alarms retained salient and attention-grabbing characteristics.

This study investigated both the subjective impressions and objective recognition performance of healthcare providers regarding the redesigned alarm sounds. Assessments were conducted with the same participants before and after the clinical introduction of the new alarms in operating rooms. We hypothesized that the refined alarm tones would be perceived as more pleasant without compromising their functional distinctiveness or recognizability.

## 2. Materials and Methods

### 2.1. Refined Alarm Sounds

We conducted a comparison between the conventional Philips alarm sounds and the newly developed refined alarm sounds by Philips to evaluate differences in perceived sound appeal, functionality, and recognizability. The refined alarm sounds were developed independently of the present study and were certified and released as a commercially available Philips product before this study took place [22]. Although the design process was not part of our investigation, its underlying goals are briefly summarized here for context and transparency.

To improve the auditory perception of patient monitoring alarm sounds, the sound designers at Sen Sound employed a human user-centered design approach. Over the course of six months, two iterative development cycles were conducted, combining qualitative methods (interviews and co-creation workshops) with quantitative surveys. More than 100 healthcare professionals, primarily anesthesia providers, including nurses, resident physicians, and attending anesthesiologists, from seven countries contributed to defining the characteristics they associated with improved alarm sounds.

Key findings indicated that a more pleasant sound quality, reduced acoustic harshness, and a more natural sonic character were considered essential for making alarms less stressful. To reduce cognitive load, alarm tones were also expected to be more clearly distinguishable across priority levels and more perceptually distinct from surrounding ambient sounds.

Low-priority alarms were redesigned with a rounder acoustic profile and longer intervals between tones. In contrast to the previous design, where pitch and interval were identical, medium-priority alarms were given distinct frequency and timing characteristics. High-priority alarms were softened in their acoustic sharpness while maintaining their attention-grabbing properties.

A direct auditory comparison of the traditional and refined alarm sounds is available in Appendix A. A detailed description of the design process can be found in Appendix A.

### 2.2. Study Design and Setting

We conducted a single-center, investigator-initiated, pre–post intervention study at the University Hospital Zurich, Switzerland. The study evaluated anesthesia providers’ perceptions of the sound appeal (i.e., listening comfort) and functionality of both the traditional and refined Philips alarm sounds, as well as their ability to correctly identify and assign alarm sounds to the appropriate priority levels. A flowchart of the study procedure is presented in Figure 1.

### 2.3. Ethical Considerations and Data Security

The study protocol was reviewed by the responsible ethics committee in Zurich, Switzerland, which issued a statement of non-jurisdiction (BASEC Number Req-2022-00689). Written informed consent was obtained from all participants prior to enrollment. Participants did not receive any compensation. All data reported in this study are anonymized, and data processing was conducted on the secure IT infrastructure of the University Hospital Zurich and the University of Zurich to ensure confidentiality and data protection. Data custody and analysis were fully under the control of the academic investigators. The manufacturer of the refined alarm system had no access to raw data and did not participate in the study design, data analysis, interpretation of results, or manuscript preparation. All analyses were conducted independently by JB from the Epidemiology, Biostatistics and Prevention Institute, University of Zurich, Switzerland, who had no involvement with or connection to the manufacturer. De-identified datasets and analytic code will be made available upon reasonable request, as stated in the Data Availability Section.

### 2.4. Study Participants

This study included anesthesia providers, comprising trainee and certified anesthesia nurses, resident physicians, and attending anesthesiologists. During the pre-intervention phase, the only inclusion criterion was regular exposure to patient monitoring systems and their associated alarm sounds; no further inclusion or exclusion criteria were applied.

Recruitment was based on participants’ availability during clinical operations. The study team approached potential participants in operating areas where visceral, thoracic, ENT, urological, gynecological, neurosurgical, and ophthalmological procedures were performed, briefly explained the study, and invited them to participate. All approached individuals agreed to take part.

For the post-intervention phase, only participants from the pre-phase were re-enrolled to allow within-subject comparisons. In January 2024, a one-hour training session was conducted to familiarize staff with the refined alarm sounds and their upcoming clinical implementation.

### 2.5. Study Procedure

The study comprised three components: a demographic survey, an alarm recognition test, and a Likert-scale questionnaire evaluating the alarm sounds.

First, participants provided demographic information. They then completed an alarm recognition test in which nine alarm sounds, three per priority level (low, medium, high)—were played in randomized order using a MacBook Air (Apple Inc., Cupertino, CA, USA). The tests were conducted in a controlled environment with standardized volume adjustment and realistic operating-room background noise. While no formal Sound Pressure Level (SPL) or frequency-response calibration was performed, prior informal listening confirmed that the acoustic quality of the MacBook Air speakers was comparable to that of clinical patient monitors.

During the pre-intervention phase, participants assessed traditional alarm sounds, whereas the post-intervention phase involved the refined Philips alarm sounds. After each sound, participants entered their perceived priority level using an online survey (Google Forms, Google LLC, Mountain View, CA, USA) on an iPad (Apple Inc., Cupertino, CA, USA).

Subsequently, participants completed a standardized questionnaire in the same survey tool. The questionnaire consisted of five items assessing perceived sound appeal and six items evaluating perceived functionality. The items were developed by an interdisciplinary expert panel comprising sound engineers, anesthesia nursing staff, physicians, and a survey methodology specialist.

Following the implementation of the refined Philips alarm sounds in April 2024, a four-month familiarization phase was allowed. This duration was chosen as a pragmatic compromise, long enough for staff to adapt to the new sounds, yet feasible within the high-stakes operating room environment. In September 2024, the entire procedure was repeated with the same participants.

To complement the evaluation, the post-intervention questionnaire included three additional statements:-I like the refined alarm sounds better.-I am open to further changes to the patient monitoring alarm sounds.-The alarm sounds could be made even more pleasant without compromising functionality.

### 2.6. Outcomes

The primary outcome of this study was the perceived sound appeal of the alarm sounds, measured as the mean score across five sub-items, each rated on a four-point Likert scale. Higher scores reflected more favorable evaluations. This outcome was chosen because the central design goal of the newly developed alarm sounds was to improve auditory comfort and thereby reduce stress for both healthcare providers and patients.

Secondary outcomes included perceived functionality and recognizability of the alarm sounds:

Perceived functionality was assessed as the mean score across six Likert-scale items evaluating how well the alarm sounds supported clinical prioritization and interpretation.

Recognizability was defined as the number of alarm sounds correctly assigned to their respective urgency levels (low, medium, high) by each participant.

In addition to the aggregated mean scores, the five individual items assessing sound appeal and the six items assessing functionality were analyzed descriptively and are presented graphically to illustrate item-level response distributions. Recognizability results are also presented graphically, displaying the proportion of correct, partially correct, and incorrect assignments for each priority level in both study phases.

### 2.7. Sample Size

Due to the nature of the study and the inability to fully replicate post-implementation conditions in a classical pilot setting, we conducted a power analysis based on perceived sound appeal ratings from the first 10 participants (internal pilot). We acknowledge that estimates of the mean difference and standard deviation from such a small sample are subject to uncertainty, and therefore the resulting sample size should be considered an approximation.

Assuming a mean Likert-scale increase of 0.3 points (equivalent to an 11.5% improvement) and an estimated standard deviation of differences of 1, the required sample size to detect this effect with 90% power at a significance level of α = 0.05 was calculated as *n* = 73. To account for potential dropouts or incomplete data, we aimed to recruit 90 participants. In addition to power considerations, we assessed the expected precision of the mean difference estimate. With 73 participants (power-based calculation), the 95% confidence interval for the mean difference is approximately ±0.23 points (0.071 to 0.529), while with 90 participants (planned recruitment), it narrows to approximately ±0.21 points (0.093 to 0.507). This precision-based framing ensures that the study results will be both statistically powered and reliably interpretable, even given the inherent variability of subjective outcomes.

### 2.8. Statistical Analysis

We report means and standard deviations (SD) for continuous variables and absolute and relative frequencies for categorical variables. As each participant completed the recognition test and questionnaire both before and after the introduction of the refined alarm sounds, measurements were not independent. Accordingly, we employed mixed-effects regression models with a random intercept per participant to account for within-subject correlation.

For the continuous outcomes perceived sound appeal (primary outcome) and perceived functionality, we used linear mixed models.

For the composite outcome perceived sound appeal, the Likert-scale responses for negatively phrased items were inverted prior to analysis to ensure consistent interpretation, with higher scores reflecting more favorable perceptions across all items.

For the count outcome number of correctly recognized alarm sounds, we applied a mixed Poisson regression model.

All models were adjusted for: age, gender, years of professional experience, musical training (binary), and participation in the one-hour training session in January 2024 (binary).

To further explore recognizability, we conducted a sub-analysis by alarm priority level (low, medium, high) using a mixed logistic regression model that included an interaction term between time (pre/post) and alarm priority. In addition, a confusion matrix was constructed to visualize misclassifications across urgency levels.

In addition to the main analyses, we conducted a sensitivity re-analysis excluding participants who reported attending the January 2024 training session. This was done to account for potential prior exposure to the refined sounds before the pre-phase. The mixed models for Perceived Sound Appeal, Perceived Functionality, and Recognizability were re-run using this restricted sample to assess the robustness of the results.

All data preparation was performed in Microsoft Excel (Microsoft Corporation, Redmond, WA, USA). Statistical analyses were conducted using R (version 4.4.2; R Foundation for Statistical Computing, Vienna, Austria). Figures were created using GraphPad Prism (version 10.3.1; GraphPad Software Inc., San Diego, CA, USA). A *p*-value of <0.05 was considered statistically significant.

## 3. Results

### 3.1. Participants

A detailed overview of participant demographics and study characteristics is provided in Table 1 and Figure 1. Between January and March 2024 (pre-intervention phase), 90 participants were recruited. In April 2024, the refined alarm sounds were implemented on patient monitors in the operating rooms of the study center. Following a four-month acclimatization period, 77 of the original 90 participants completed the post-intervention phase between August and October 2024. Reasons for participant dropout are outlined in Figure 1.

### 3.2. Perceived Sound Appeal

Before reporting pre- and post-phase results, we evaluated the psychometric properties of the Perceived Sound Appeal scale. The internal consistency of the scale was moderate to low (coefficient ω = 0.489). Exploratory factor analysis (EFA) suggested a single-factor structure, although some items showed high uniqueness, indicating that the scale may capture broader or more heterogeneous aspects of sound appeal, potentially reflecting differences in individual understanding or preferences (see Appendix A).

In the pre-phase, the mean Likert-scale score for perceived sound appeal was 2.27 (SD 0.41), compared to 2.78 (SD 0.46) in the post-phase. The linear mixed regression model revealed very strong evidence for an increase of 0.51 points (95% CI: 0.37–0.64; *p* < 0.001) following the implementation of the refined alarm sounds. This indicates that, on average, the refined sounds were rated 0.5 points higher on a four-point scale, reflecting a substantially more favorable perception.

Notably, the Likert-scale ratings indicated a marked improvement in the perceived patient experience, with users reporting that the refined alarm sounds appeared less distressing for patients.

No evidence for associations was observed between the outcome and age, gender, years of experience, musical background, or participation in the training session. A graphical summary for the five individual items contributing to the sound appeal score is shown in Figure 2A.

### 3.3. Perceived Functionality

Prior to examining pre- and post-phase results, we assessed the psychometric properties of the Perceived Functionality scale. The scale demonstrated moderate internal consistency (coefficient ω = 0.652). Exploratory factor analysis indicated a primarily single-factor structure, with most items exhibiting strong loadings, although a few items showed higher uniqueness, suggesting that certain items may capture slightly different facets of functionality perception (see Appendix A).

The mean Likert-scale score for perceived functionality was 3.00 (SD 0.39) in the pre-phase and 2.85 (SD 0.42) in the post-phase. The linear mixed regression model showed a small decrease in perceived functionality for the refined Philips alarm sounds compared to the traditional sounds, with a coefficient of −0.15 (95% CI: −0.27 to −0.03).

A pronounced change was observed in the Likert scale measuring motivational impact, with participants indicating that the refined alarm sounds increased their motivation to attend to the monitor. Conversely, perceived discriminability between alarm tones was lower in the post-intervention phase.

None of the covariates included in the model (age, gender, years of experience, musical background, or participation in the training) showed evidence for an effect (*p* > 0.05 for all). The item-level responses for perceived functionality are visualized in Figure 2B.

### 3.4. Recognizability

In the recognition test, participants were asked to assign nine alarm sounds (three per priority level) to their corresponding urgency categories. Recognition performance was slightly higher in the pre-intervention phase (mean = 7.73, SD = 1.81) compared to the post-intervention phase (mean = 7.14, SD = 1.72). The mixed Poisson regression model yielded a rate ratio of 0.93 (95% CI: 0.83–1.04). This suggests that the rate in the intervention group was approximately 7% lower compared to the control group. However, the 95% confidence interval includes the null value of 1.0, indicating that the true effect could range from a 17% reduction to a 4% increase in the rate.

A sub-analysis was conducted to evaluate recognizability by individual alarm priority levels:-Low-priority: 95.2% (220/231) correct in pre-phase vs. 87.5% (202/231) in post-phase.-Medium-priority: 81.1% (187/231) correct vs. 62.0% (143/231).-High-priority: 81.4% (188/231) vs. 89.0% (205/231) with refined sounds.-Mixed logistic regression results:-Low-priority alarms: OR 0.29 (95% CI: 0.14–0.63; *p* = 0.002).-Medium-priority alarms: OR 0.32 (95% CI: 0.19–0.51; *p* < 0.001).-High-priority alarms: OR 2.05 (95% CI: 1.15–3.67; *p* = 0.002).

These findings are visualized in Figure 3A. A confusion matrix (Figure 3B) shows that medium-priority alarms in the post-phase were more frequently misclassified as low-priority.

### 3.5. Sensitivity Analysis

To address potential bias from prior exposure to the refined alarm sounds, a sensitivity re-analysis was conducted excluding participants who reported attending the January 2024 training session. Two participants with inconsistent responses regarding training attendance between the pre- and post-phase were also excluded.

The mixed Poisson and linear mixed regression models for Perceived Sound Appeal, Perceived Functionality, and Recognizability were re-estimated using this restricted sample. Across all three outcomes, the results were consistent with those of the main analyses, indicating that exclusion of trained participants did not materially affect the findings. The effect sizes and confidence intervals remained comparable, confirming the robustness of the observed improvements following the implementation of the refined alarm sounds. Detailed results from the sensitivity analyses are presented in Appendix A.

### 3.6. Perceptions and Preferences Regarding the Refined Alarm Sounds

71% (55/77) of participants preferred the refined alarm sounds.

92% (71/77) were open to further changes in patient monitoring alarms.

55% (42/77) expressed concern that making alarms more pleasant could compromise functionality.

## 4. Discussion

This pre–post intervention study evaluated the impact of redesigned patient monitoring alarm sounds following real-life implementation in a university hospital setting. The refined sounds were specifically developed to reduce acoustic harshness by addressing features such as high pitch, bright timbre, sharp amplitude envelopes, and short repetition intervals, to enhance user experience.

Our findings suggest that anesthesia providers experienced the redesigned alarm sounds as significantly more pleasant, although they also perceived them as slightly less functional. Recognition accuracy showed mixed results: low- and medium-priority alarms were better recognized with the traditional sounds, whereas high-priority alarms were identified more accurately using the refined tones. Overall, the redesigned alarms appear to improve the perceived auditory experience but may involve a modest trade-off in functional clarity and recognizability.

Overall, most participants preferred the new alarm sounds and expressed openness to further improvements in alarm design.

To interpret these results, it is important to consider the dual role of acoustically harsh sounds. On one hand, auditory features associated with harshness, such as high pitch, abrupt onsets, and bright timbre, effectively convey urgency and are rapidly processed by the brain, making them well-suited for high-priority alarms [24,25,26]. On the other hand, such sounds also activate neural circuits associated with stress and aversion [27,28], contributing to alarm fatigue and discomfort in clinical environments.

To address this, the redesign focused on reducing acoustic harshness in low- and medium-priority alarms to improve overall auditory comfort, while maintaining the urgency and salience of high-priority alarms.

Our results suggest that this trade-off was partially successful: the perceptual contrast between alarm levels was enhanced, leading to improved recognition of critical (high priority) alerts. However, this improvement came at the cost of reduced discriminability between low- and medium-priority tones, as medium-priority alarms were more frequently misclassified as low-priority in the post-intervention phase, showing a usability safety trade-off. At the same time, recognition of high-priority alarms improved, suggesting that the enhanced perceptual contrast successfully supported the identification of the most critical alerts.

This finding underscores the importance of a thoughtful and deliberate use of harsh sound attributes in medical alarm design [6,7]. Given that most medical device alarms do not require immediate action [5], it is reasonable to reduce acoustic harshness in non-critical low- and medium-priority alarms to alleviate stress among healthcare providers and patients. Conversely, reserving the most alerting auditory features for high-priority alarms can enhance their perceptual distinctiveness and improve clinical relevance [6,7]. From a risk management perspective, the observed misclassification of some medium-priority alarms highlights the need for interim mitigations when implementing refined sound designs. In clinical practice, the risk is mitigated by concurrent visual cues—color coding (blue, yellow, red) and textual labels—which provide clear and redundant information about alarm priority. Moreover, the low-priority alarm type in this study primarily represents technical alerts rather than direct patient risk, further reducing safety concerns. Nevertheless, accompanying staff education and clear user guidance should be considered in future implementations to ensure that auditory refinements do not compromise situational awareness.

Even if it appears clinically sensible that the most critical alarms should be most easily recognizable, there remains a safety trade-off to consider for low- and medium-priority alarms. The update of the Philips monitoring alarm sounds represents a promising step toward reducing stress associated with the outdated auditory environment of conventional alarm systems. While this refinement addresses specific issues such as acoustic harshness and the overall unpleasantness of clinical soundscapes, it stops short of incorporating more innovative paradigms, such as auditory icons or speech-based alarms, which have been shown to improve perceptual clarity and situation awareness in complex environments [14,15,29]. Had such approaches been considered, it is possible that the user experience could have been further enhanced without compromising the recognition and functionality of low- and medium-priority alarms, thereby mitigating the safety trade-off observed in our study. Notably, the international standard IEC 60601-1-8 explicitly permits more intuitive alarm formats, including auditory icons.

An important point for the further development of acoustic alarm systems is how new alarm designs are integrated into the existing hospital soundscape. Even if individual alarms are easily recognized and distinguishable in isolation, they may be misinterpreted or masked in the presence of loud background noise or other simultaneous acoustic alarms, potentially reducing their effectiveness in clinical environments. This highlights the need for considering cross-platform alarm design, ensuring that alarm sounds remain consistent and interpretable across different devices and monitor systems, which could support broader standardization and improve overall patient safety.

Rather than a fundamental redesign, the refined Philips alarm sounds may be seen as an incremental improvement, aimed at addressing well-known shortcomings while preserving the familiar beep-based architecture. In this light, the present update may represent a transitional step toward a more user-centered and cognitively ergonomic acoustic environment in the acute care domain.

This study has both strengths and limitations. As with all pre–post designs, time-related trends or external influences may have affected the observed outcomes. However, a key strength of the pre–post intervention design is its ability to directly compare outcomes within the same population, enhancing internal validity and enabling a real-world evaluation of the intervention’s impact [30,31]. We acknowledge that if these acoustic alarm systems are to be further developed and optimized, structured longitudinal monitoring with iterative feedback would be necessary, an aspect that our study does not include. Another limitation related to the pre–post design is that some participants had been exposed to the refined alarm sounds during a training session prior to the pre-phase of the study. However, a reanalysis including only participants without any prior exposure to the refined sounds revealed no differences compared with the overall results.

Regarding the Likert scales used, both the Perceived Sound Appeal and Perceived Functionality scales were newly developed for this study and should be considered exploratory. The limited number of items, heterogeneous item formulations, and moderate to low internal consistency suggest that these scales capture broad but partly overlapping aspects of the intended constructs. Some items also showed high uniqueness, indicating that they may reflect distinct subdimensions rather than a single underlying factor. Further validation using larger samples, test–retest reliability, and item response or ordinal modeling is warranted.

A further limitation concerns the playback setup. The alarm sounds were presented via MacBook Air built-in speakers without formal sound pressure level calibration or frequency-response matching to the clinical monitor speakers. Although all participants used the same device model under identical conditions and could adjust playback volume to a comfortable level, this setup may not fully replicate the acoustic properties of clinical environments. Informal listening comparisons suggested that the MacBook speakers reproduced the alarm tones with a sound character broadly similar to clinical monitors. Furthermore, the recognition tests were conducted under controlled conditions and therefore do not fully reflect the noise environment of an active operating room.

Another limitation is the single center setting and the restriction of the intervention to operating rooms. As a result, generalizability to other clinical environments, such as intensive care units or emergency departments, may be limited [32]. ICUs and similar high-noise environments are characterized by higher and more variable background noise, which could influence the perception of alarm sounds. However, single-center studies benefit from consistent institutional practices, which reduce variability and thereby enhance internal validity compared to multicenter designs [33].

Recruitment via convenience sampling may have introduced selection bias. However, we included nearly all anesthesia staff in the participating operating rooms, making it unlikely that selection bias had a substantial impact on our findings.

Finally, given that this intervention targeted a core aspect of the human–machine interface, namely, the auditory experience, its positive reception by users suggests that similar benefits may be achievable in other care settings and even among patients exposed to these alarm sounds, as suggested by Litton et al. [12].

The primary outcome of our study was perceived sound appeal, in line with the main goal of refining the alarm sounds. Although clinical outcomes would offer more robust evidence of impact, these measures are particularly challenging to capture in anesthesiology, where adverse events are fortunately rare and multiple factors beyond the patient monitor alarms contribute to outcomes.

## 5. Conclusions

In conclusion, reducing the harshness of patient monitoring alarm sounds improved the auditory experience for anesthesia providers, making the alarms more pleasant to listen to. However, this modification was associated with a slight decrease in perceived functionality and mixed effects on recognition accuracy: low- and medium-priority alarms were less reliably identified, whereas high-priority alarms were more easily distinguished. These findings highlight a safety-relevant trade-off, in which enhanced sound appeal comes at the cost of somewhat reduced functional clarity and recognizability for certain alarm priorities.

While the refined Philips alarm sounds may contribute to a less harsh hospital soundscape, further improvements will likely require the adoption of more innovative approaches. Concepts such as auditory icons and voice-based alarms have shown promising potential in experimental research. The alarm sounds evaluated in this study could serve as a preliminary reference or complement within such advanced systems, which aim to support situation awareness and user acceptance in clinical practice.

To fully understand their impact and generalizability, future studies should also investigate the effectiveness and acceptance of these sounds in other clinical environments, such as intensive care units and emergency departments, and among patients [14,15,29].

## Figures and Tables

**Figure 1 healthcare-13-03033-f001:**
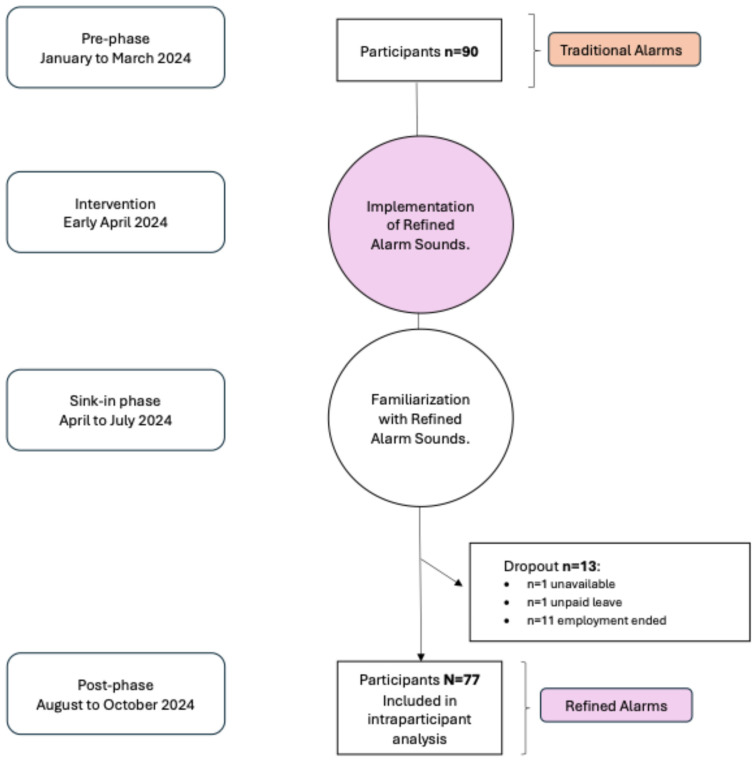
Flow diagram illustrating the timeline and participant flow of the pre–post intervention study.

**Figure 2 healthcare-13-03033-f002:**
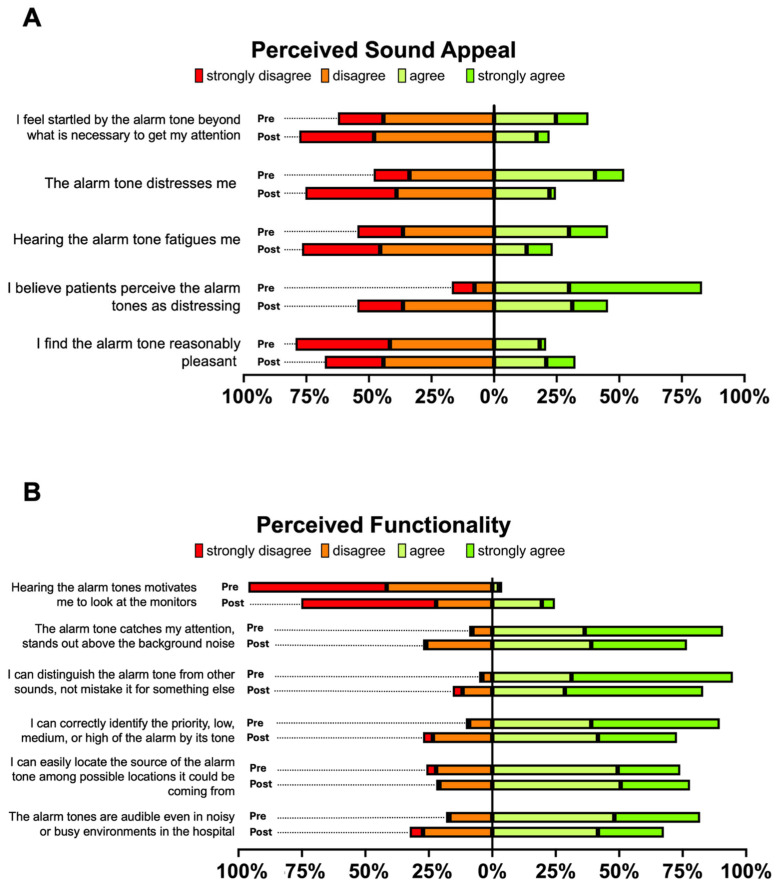
Perceived Sound Appeal (**A**) and Perceived Functionality (**B**) of Alarm Sounds Before and After the Intervention. Stacked bar charts illustrating participants’ responses to five items assessing perceived sound appeal (Panel **A**) and six items assessing perceived functionality (Panel **B**) of alarm sounds, each rated on a four-point Likert scale (strongly disagree to strongly agree). Results are shown separately for the pre-intervention (traditional alarm sounds) and post-intervention (refined Philips alarm sounds) phases.

**Figure 3 healthcare-13-03033-f003:**
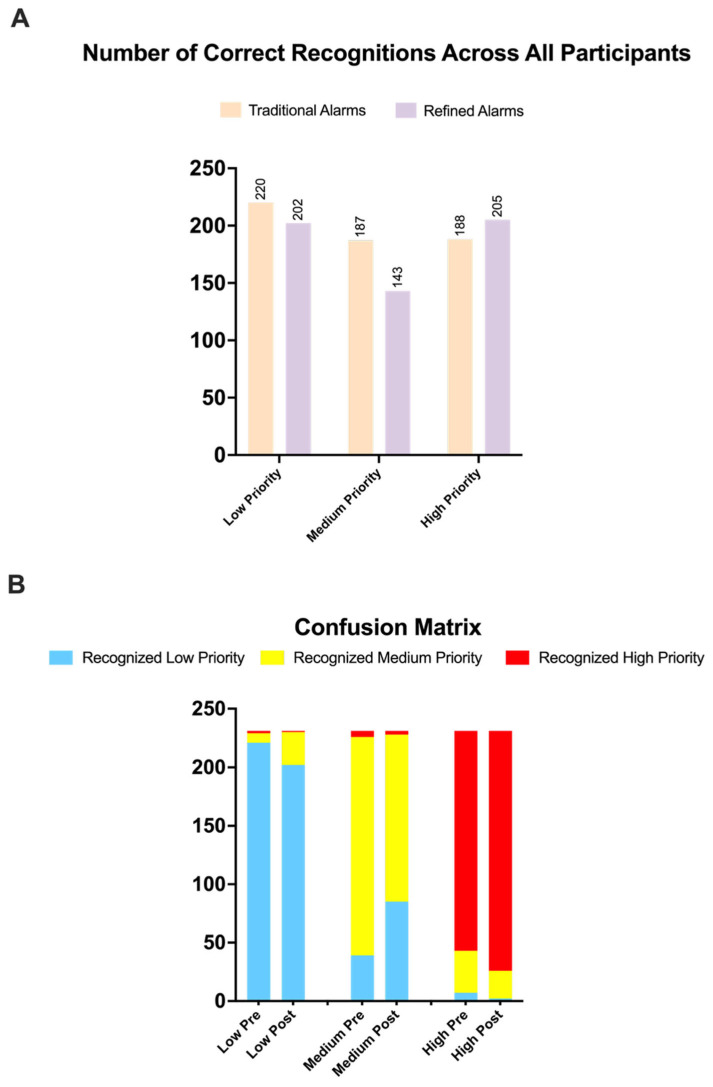
Recognition Accuracy and Misclassification Patterns of Alarm Priorities Before and After Implementation of Refined Alarm Sounds (**A**): Bar charts showing the proportion of correctly recognized alarm sounds across the three priority levels (low, medium, high) in the pre- and post-intervention phases. (**B**): Confusion matrix visualizing misclassification patterns by alarm priority and study phase. Each stacked bar represents how frequently sounds from one priority level were assigned to each of the three categories. Notably, medium-priority alarms in the post-phase were often misclassified as low-priority, while the distinction of high-priority alarms improved.

**Table 1 healthcare-13-03033-t001:** Participant and study characteristics. Values are presented as numbers, proportions (%), or medians (range).

Participants used for analysis; N	77
Participants in pre-phase; *n*	90
Participants in post-phase; *n*	77
Age; y	34 (23–61)
Job experience; y	5 (0–40)
Gender; female	34 of 77 (44.2%)
Playing a musical instrument; yes	10 of 77 (13%)
Participation in the training lecture; yes	34 of 77 (44.2%)
Participation in the training lecture; no	41 of 77 (53.2%)
Participation in the training lecture; not remembered	2 of 77 (2.6%)
Job title	
Trainee nurse; *n*	4 of 77 (5.2%)
Nurse; *n*	20 of 77 (26%)
Resident physician; *n*	33 of 77 (42.9%)
Attending physician; *n*	20 of 77 (26%)

## Data Availability

The raw data supporting the conclusions of this article will be made available by the authors on request.

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
