# Peer review of "User-Centered Redesign of Monitoring Alarms: A Pre–Post Study on Perception, Functionality, and Recognizability Following Real-Life Clinical Implementation"

_healthcare, 2025, doi:10.3390/healthcare13233033_

Round 1
Reviewer 1 Report
Comments and Suggestions for Authors
Dear authors,
Please, see few comments below you might consider or clarify:
The Methods report a one-hour training session in January 2024 to familiarize staff with the refined sounds before implementation; the pre-phase also ran January–March 2024, meaning participants likely entered the pre-phase already exposed to the refined tones. This undermines internal validity of all pre–post comparisons (see Figure 1 timeline and training description).
Required fixes:
Re-analyse with a strict sensitivity set excluding anyone who attended the training (you collected this covariate) and report whether primary/secondary outcomes hold.
Explicitly acknowledge this as a design limitation and temper all causal language (e.g., “led to” → “was associated with”), including Abstract and Conclusions.
Outcome hierarchy vs safety signals: conclusions over-state “preserved functionality”
Item-level results show reduced discriminability post-implementation, and the recognizability analysis is inconclusive for non-inferiority overall, with significant drops for low/medium priority (and improvement for high). Yet the conclusion emphasises preserved functionality. This is inconsistent with the data and could be read as minimising a safety trade-off.
Required fixes:
Reframe the Abstract and main Conclusions to foreground the safety-relevant trade-off: improved appeal and high-priority recognition at the cost of lower low/medium differentiation; avoid implying overall functionality is “preserved” without qualification.
Add a risk-management paragraph: what interim mitigations (e.g., staff education, visual cues) are needed given the medium-priority misclassification seen in Figure 3B?
Recognition tests were played through MacBook speakers, with only a qualitative statement that acoustic quality was “comparable” to monitor speakers; no SPL/loudness calibration or frequency-response matching is provided. This threatens both construct and external validity of recognizability results.
Required fixes:
Describe and document calibration (SPL at participant ear, frequency-response matching), or repeat the recognition test with monitor playback (or calibrated headphones) and report whether the findings replicate.
Non-inferiority margins (–0.5 Likert points; RR 0.90) are author-defined post hoc “based on clinical expertise” with no external justification or registration; the recognizability result is inconclusive against that margin.
Required fixes:
Provide a priori justification (literature/clinician consensus) and sensitivity analyses across plausible margins; otherwise, avoid non-inferiority language and present results descriptively with CIs.
The five “appeal” and six “functionality” items were expert-devised but no reliability (α/ω), dimensionality, or test-retest evidence is given. Aggregating Likert items into means without validation risks measurement artefacts.
Required fixes:
Report internal consistency, item–total correlations, and a brief factor analysis (or confirmatory model) for both scales in the Supplement; consider IRT or at minimum ordinal models for item analysis.
The refined alarms are a commercial product; the manufacturer assisted activation; two coauthors (vendor) were compensated; and several authors have broader manufacturer-related IP/financial ties (albeit not to the specific alarm sounds). Current safeguards are not described beyond disclosure.
Required fixes:
Clarify data custody, analytic independence, and any manufacturer role in design/analysis/interpretation; ideally add an independent replication of the analysis (blinded to phase) and deposit de-identified data/code upon request (per your Data Availability statement).
Single-centre OR-only, no concurrent control: Limits generalizability; expand on time-trend confounding inherent to pre–post designs and justify the 4-month familiarization choice.
Sample size planning: Based on the first 10 participants (internal pilot) for a subjective outcome; discuss implications and consider precision-based framing.
Stimulus set limitation: Only nine sounds (three per priority) limits per-tone inference; provide per-tone performance (item analysis) and a confusion matrix per tone in the Supplement.
Ecological noise: The Discussion notes masking/ambient noise issues, yet recognition tests were not run under realistic noise; add this as a limitation and, if feasible, a follow-up test in representative OR noise.
Best wishesAuthor Response
Dear Reviewer 1,
Point by Point Response
We would like to sincerely thank the three reviewers and the editor for their time and the intellectual effort they dedicated to our paper. We have addressed each comment below with great care and believe that the manuscript has become substantially more transparent and engaging for the reader as a result.
Reviewer 1
Comment 1: The Methods report a one-hour training session in January 2024 to familiarize staff with the refined sounds before implementation; the pre-phase also ran January–March 2024, meaning participants likely entered the pre-phase already exposed to the refined tones. This undermines internal validity of all pre–post comparisons (see Figure 1 timeline and training description).
Required fixes: Re-analyse with a strict sensitivity set excluding anyone who attended the training (you collected this covariate) and report whether primary/secondary outcomes hold. Explicitly acknowledge this as a design limitation and temper all causal language (e.g., “led to” → “was associated with”), including Abstract and Conclusions.
Response 1: In response to the reviewer’s comment regarding potential exposure to the refined sounds prior to the pre-phase, we conducted a sensitivity analysis restricted to participants who did not attend the January 2024 training lecture. Attendance was assessed via self-report in the survey. Two participants indicated “yes” to attending the training in the pre-phase but “no” in the post-phase; these participants were excluded from the sensitivity analysis to ensure data consistency.
The mixed models for Perceived Sound Appeal, Perceived Functionality, and Recognizability were re-estimated using this restricted sample. The results did not differ meaningfully from the analyses based on the full study population, indicating that exclusion of trained participants did not affect the overall findings. Both the re-analysis and the initial models are presented in Multimedia Supplement 2. In addition, we have updated the Methods (page 7, line 284 ff.) and Results (page 11, line 381 ff.) to include the sensitivity re-analysis excluding participants who attended the January 2024 training session. We also mentioned this in the limitations (page 13 and line 436 - 439).
Comment 2: Outcome hierarchy vs safety signals: conclusions over-state “preserved functionality. Item-level results show reduced discriminability post-implementation, and the recognizability analysis is inconclusive for non-inferiority overall, with significant drops for low/medium priority (and improvement for high). Yet the conclusion emphasises preserved functionality. This is inconsistent with the data and could be read as minimising a safety trade-off.
Required fixes: Reframe the Abstract and main Conclusions to foreground the safety-relevant trade-off: improved appeal and high-priority recognition at the cost of lower low/medium differentiation; avoid implying overall functionality is “preserved” without qualification. Add a risk-management paragraph: what interim mitigations (e.g., staff education, visual cues) are needed given the medium-priority misclassification seen in Figure 3B?
Response 2: We thank the reviewer for this observation. In response, we have revised the Abstract (page 1, line 27 and 37 to 39) and Conclusion (page 14, line 521 ff.) to more accurately reflect the results and to acknowledge the safety-relevant trade-off between improved sound appeal and reduced discriminability for low- and medium-priority alarms. We no longer describe functionality as “preserved,” but instead note that perceived functionality was slightly lower and that recognition accuracy showed mixed results. We also refrain from using non-inferior language, as requested in comment 5. Furthermore, we have added a paragraph in the Discussion (page 13, line 437 ff.) addressing risk management considerations. Specifically, we now describe that, in the current clinical context, the risk associated with misclassification of medium-priority alarms is mitigated by concurrent visual cues (color coding and textual labels) and by the non-critical nature of low-priority technical alarms. We also note that future implementation should include staff education and user guidance to maintain situational awareness when adopting refined alarm designs.
Comment 3: Recognition tests were played through MacBook speakers, with only a qualitative statement that acoustic quality was “comparable” to monitor speakers; no SPL/loudness calibration or frequency-response matching is provided. This threatens both construct and external validity of recognizability results.
Required fixes: Describe and document calibration (SPL at participant ear, frequency-response matching), or repeat the recognition test with monitor playback (or calibrated headphones) and report whether the findings replicate.
Response 3: We thank the reviewer for this valuable comment and fully acknowledge the limitation regarding playback calibration. The recognition tests were conducted using MacBook Air built-in speakers. While no formal SPL or frequency-response calibration was performed, several measures were taken to ensure a consistent and realistic auditory experience:
- Controlled playback environment: All tests were conducted in a quiet room using the same device model and sound output settings for every participant, ensuring consistent playback conditions across sessions.
- Volume adjustment procedure: A standardized test sound was provided at the beginning of the session to allow participants to adjust the playback level to a comfortable listening volume, reducing interindividual variability in perceived loudness.
- Ecological validity: The playback stimuli consisted of the alarm tones mixed with authentic operating-room background noise, which was intended to reflect realistic clinical listening conditions rather than precise acoustic calibration.
- Comparability to clinical playback: Although no formal acoustic calibration was performed, informal listening tests indicated that the MacBook Air speakers reproduced the alarm tones with a sound character subjectively similar to the clinical monitor loudspeakers, particularly in the mid-frequency range where alarm tones predominantly occur.
We acknowledge that the lack of formal SPL and frequency-response calibration limits strict comparability to real monitor playback and thus represents a methodological constraint. However, given that all participants were tested under identical conditions and the observed effects were large and consistent, we believe that the overall pattern of results remains robust. Nevertheless, we have included this point in the limitations of the study (page 14, line 492 ff.).
Comment 4: Non-inferiority margins (–0.5 Likert points; RR 0.90) are author-defined post hoc “based on clinical expertise” with no external justification or registration; the recognizability result is inconclusive against that margin.
Required fixes: Provide a priori justification (literature/clinician consensus) and sensitivity analyses across plausible margins; otherwise, avoid non-inferiority language and present results descriptively with CIs.
Response 4: We thank the reviewer for this valuable comment. In response, we have removed all non-inferiority terminology and the predefined margins from the manuscript. Initially, we considered a margin corresponding to approximately 10% lower perceived functionality (–0.5 Likert points; relative risk 0.90) as the threshold for clinical relevance, based on expert consensus within the study team. However, as there are no established or validated non-inferiority margins for this type of perceptual experiment, we agree that such an approach is not sufficiently justified. We therefore chose to present the results descriptively, reporting model coefficients and 95% confidence intervals without non-inferiority claims. The Methods, Results, Discussion, and Conclusion sections have been revised accordingly to reflect this more cautious interpretation.
Comment 5: The five “appeal” and six “functionality” items were expert-devised but no reliability (α/ω), dimensionality, or test-retest evidence is given. Aggregating Likert items into means without validation risks measurement artefacts.
Required fixes: Report internal consistency, item–total correlations, and a brief factor analysis (or confirmatory model) for both scales in the Supplement; consider IRT or at minimum ordinal models for item analysis.
Response 5: We thank the reviewer for this important point. In response, we conducted psychometric analyses to evaluate internal consistency and dimensionality of the scales. These analyses indicate that the Perceived Functionality scale has moderate internal consistency and largely loads onto a single factor, while the Perceived Sound Appeal scale shows lower internal consistency and some items with high uniqueness. This suggests that the Perceived Sound Appeal items may capture broader or more heterogeneous aspects of sound appeal, potentially reflecting differences in individual understanding or preferences, for example, varying musical tastes such as classical versus heavy metal music. The lower internal consistency may also reflect that the items are formulated differently and cover distinct aspects of sound appeal.
We acknowledge that these scales are preliminary and have added the results to the Supplementary Materials. Future work could employ item response theory or ordinal modeling to further validate these measures. We bring the analyses in the results (page 8; lines 306 to 312 and lines 327 to 332) and have listed it explicitly under limitations (page 13; lines 484 ff.).
Analysis performed using JASP version 0.95.4 (Apple Silicon)
Perceived Functionality
|
Frequentist Scale Reliability Statistics |
||||
|
95% CI |
||||
|
Coefficient |
Estimate |
Std. Error |
Lower |
Upper |
|
Coefficient ω |
0.652 |
0.041 |
0.571 |
0.733 |
|
Note. The following item correlated negatively with the scale: Functionality 1. |
||||
|
Frequentist Individual Item Reliability Statistics |
||||
|
Coefficient ω (if item dropped) |
||||
|
Item |
Estimate |
Lower 95% CI |
Upper 95% CI |
|
|
Functionality2 |
0.500 |
0.387 |
0.613 |
|
|
Functionality3 |
0.509 |
0.397 |
0.620 |
|
|
Functionality4 |
0.616 |
0.527 |
0.706 |
|
|
Functionality5 |
0.610 |
0.519 |
0.701 |
|
|
Functionality6 |
0.536 |
0.431 |
0.641 |
|
|
Functionality 1 |
0.787 |
0.734 |
0.840 |
|
Exploratory factor analysis
|
Factor Loadings |
||
|
|
Factor 1 |
Uniqueness |
|
Functionality2 |
0.843 |
0.290 |
|
Functionality3 |
0.842 |
0.290 |
|
Functionality6 |
0.801 |
0.359 |
|
Functionality 1 |
0.748 |
0.440 |
|
Functionality5 |
0.550 |
0.698 |
|
Functionality4 |
0.539 |
0.710 |
|
Note. Applied rotation method is promax. |
||
Perceived Sound Appeal
|
Frequentist Scale Reliability Statistics |
||||
|
95% CI |
||||
|
Coefficient |
Estimate |
Std. Error |
Lower |
Upper |
|
Coefficient ω |
0.489 |
0.058 |
0.375 |
0.603 |
|
Note. The following item correlated negatively with the scale: Sensibility5. |
||||
|
Frequentist Individual Item Reliability Statistics |
||||
|
Coefficient ω (if item dropped) |
||||
|
Item |
Estimate |
Lower 95% CI |
Upper 95% CI |
|
|
Sensibility1 |
0.400 |
0.266 |
0.534 |
|
|
Sensibility2 |
0.260 |
0.072 |
0.447 |
|
|
Sensibility3 |
0.334 |
0.182 |
0.485 |
|
|
Sensibility4 |
0.523 |
0.412 |
0.635 |
|
|
Sensibility5 |
0.622 |
0.531 |
0.714 |
|
Exploratory factor analysis
|
Factor Loadings |
||
|
|
Factor 1 |
Uniqueness |
|
Sensibility2 |
0.845 |
0.286 |
|
Sensibility3 |
0.709 |
0.498 |
|
Sensibility1 |
0.537 |
0.712 |
|
Sensibility4 |
|
0.959 |
|
Sensibility5 |
|
0.880 |
|
Note. Applied rotation method is promax. |
||
Comment 6: The refined alarms are a commercial product; the manufacturer assisted activation; two coauthors (vendor) were compensated; and several authors have broader manufacturer-related IP/financial ties (albeit not to the specific alarm sounds). Current safeguards are not described beyond disclosure.
Required fixes: Clarify data custody, analytic independence, and any manufacturer role in design/analysis/interpretation; ideally add an independent replication of the analysis (blinded to phase) and deposit de-identified data/code upon request (per your Data Availability statement).
Response 6: We thank the reviewer for this important comment. We have expanded the Ethical Considerations and Data Security section (page 4; lines 171 ff.) of the Methods to clarify data custody, analytic independence, and the manufacturer’s role. Specifically, data custody and analysis were fully under the control of the academic investigators. The analyses were conducted independently by Julia Braun (Epidemiology, Biostatistics and Prevention Institute, University of Zurich), who had no involvement with or connection to Philips.Representatives from Philips were allowed to read the manuscript prior to submission, but they had no access to raw data and were not able to make any changes or influence the study design, data analysis, interpretation of results, or manuscript preparation. Yoko and Avery Senn were compensated by Philips for their contributions to the development of the alarm sounds, but not for the conduct of the present study. An independent re-analysis by a third institution has not been performed. De-identified datasets and analytic code will be made available upon reasonable request, as stated in the Data Availability section.
Comment 7: Single-centre OR-only, no concurrent control: Limits generalizability; expand on time-trend confounding inherent to pre–post designs and justify the 4-month familiarization choice.
Response 7: The four-month familiarization period was necessary because another study was being conducted in the operating room during that time; otherwise, a shorter period would have been chosen. In the limitations section, we have explicitly addressed potential time-related trends and external influences inherent to the pre–post study design. Additionally, we have discussed your comments regarding the single-center setting and the fact that the study was conducted exclusively in the operating room. (pre–post study design: page 13; lines 473 ff.; single-center setting: page 14; lines 501 ff.)
Comment 8: Sample size planning: Based on the first 10 participants (internal pilot) for a subjective outcome; discuss implications and consider precision-based framing.
Response 8: Thank you very much for this valuable contribution. You are right that precision-based framing is particularly useful for subjective results where there is considerable variation and significance alone is often misleading. We have added this under Methods 2.7 sample size (page 6; lines 256 ff.).
Comment 9: Stimulus set limitation: Only nine sounds (three per priority) limits per-tone inference; provide per-tone performance (item analysis) and a confusion matrix per tone in the Supplement.
Response 9: We are not entirely sure we have understood your criticism correctly, so we would like to clarify the design of our stimulus set in more detail. In our study, three alarm sounds were tested, corresponding to low, medium, and high priority. The test set consisted of a total of nine presentations, with each priority level presented three times (3× low, 3× medium, 3× high) in random order. Therefore, each participant heard each alarm tone exactly three times, allowing for performance analysis per sound. This information is already presented in the text (page 5; lines 199 ff.) and in Figure 3a, while Figure 3b shows a confusion matrix for each alarm sound. We deliberately limited the number of repetitions to avoid overburdening participants with multiple interpretations of the same alarm sound.
Comment 10: Ecological noise: The Discussion notes masking/ambient noise issues, yet recognition tests were not run under realistic noise; add this as a limitation and, if feasible, a follow-up test in representative OR noise.
Response 10: Unfortunately, we cannot find the passage you refer to in the discussion. You are right that we conducted the recognition test in a quiet room, which does not correspond to the noise level of an active operating room. However, we played the alarm sounds with corresponding ambient sound (recorded from an operating room) (supplement 1) in order to ensure conditions that were as realistic as possible in this computer-based part of the study. Nevertheless, we have included this point under limitation (page 14; lines 494 ff.).
Reviewer 2 Report
Comments and Suggestions for Authors
I have finished my review on this single-center, pre–post intervention study evaluating a user-centered redesign of auditory alarms in clinical monitors, conducted in an operating room setting. It investigates how refined alarm sounds affect perceived sound appeal, functionality, and recognizability among anesthesia providers. It aligns well with Healthcare’s aims.
The intervention stems from a commercial collaboration and authors clearly disclose.
laptop speakers were used, is it fully replicable? is it compatible witj the acoustic fidelity of clinical monitors and when more than one patient´s monitor may be sounding.
In methods, it is mentioned how playback hardware was calibrated and comparable to clinical monitors (could be emphasized and specified).
Please define “refined” or “traditional” alarms early in Methods for clarity.
Recomendations: Please explain or discuss applicability to ICUs and other high-noise environments
Industry participation can be further explained
Studies show that in various departments equipment may not be used if no training is provided, are you planning to develop trainings? in situ? online?
Provide contextualization of use, define settings.
Author Response
Dear Reviewer 2,
Point by Point Response
We would like to sincerely thank the three reviewers and the editor for their time and the intellectual effort they dedicated to our paper. We have addressed each comment below with great care and believe that the manuscript has become substantially more transparent and engaging for the reader as a result.
Reviewer 2
Comment 1: I have finished my review on this single-center, pre–post intervention study evaluating a user-centered redesign of auditory alarms in clinical monitors, conducted in an operating room setting. It investigates how refined alarm sounds affect perceived sound appeal, functionality, and recognizability among anesthesia providers. It aligns well with Healthcare’s aims.
Response 1: We sincerely thank the reviewer for taking the time to carefully evaluate our manuscript and for providing constructive feedback. We greatly appreciate your thoughtful comments and positive assessment of the study’s relevance to Healthcare’s aims.
Comment 2: The intervention stems from a commercial collaboration and authors clearly disclose.
Response 2: We would like to clarify that representatives from Philips were allowed to read the manuscript prior to submission; however, they had no access to the raw data and did not influence the study design, data analysis, interpretation of results, or manuscript preparation. Additionally, Yoko and Avery Senn were compensated by Philips for their contributions to the development of the alarm sounds, but not for the conduct of the present study. All further potential conflicts of interest of the academic authors have been clearly disclosed under the appropriate section. See also response to reviewer 1 comment 6.
Comment 3: laptop speakers were used, is it fully replicable? is it compatible witj the acoustic fidelity of clinical monitors and when more than one patient´s monitor may be sounding.
Response 3: We thank the reviewer for this valuable comment and fully acknowledge the limitation regarding playback calibration. As detailed in our response to Reviewer 1, comment 3, the recognition tests were conducted using MacBook Air built-in speakers in a controlled environment with standardized volume adjustment and realistic operating-room background noise. While no formal SPL or frequency-response calibration was performed, informal listening suggested a sound character similar to clinical monitors. We acknowledge that this limits strict acoustic comparability, but given identical conditions for all participants and the robust observed effects, we believe the results remain valid. This limitation has been added to the manuscript (see page 14, line 492 ff.).
Comment 4: In methods, it is mentioned how playback hardware was calibrated and comparable to clinical monitors (could be emphasized and specified).
Response 4: We have accordingly expanded the Methods section (see page 5, line 199 ff.) to include this information.
Comment 5: Please define “refined” or “traditional” alarms early in Methods for clarity.
Response 5: We have now moved the “Refined Alarm Sounds” section to the beginning of the Methods. (see page 2, line 129 ff.).
Comment 6: Recomendations: Please explain or discuss applicability to ICUs and other high-noise environments
Response 6: We thank the reviewer for this valuable suggestion. In the Limitations section, we have now addressed the partial generalizability of our results and specifically discuss applicability to intensive care units (ICUs) and other high-noise environments. We note that while our study was conducted in an operating room setting, which represents a relatively controlled acoustic environment, ICUs are characterized by higher and more variable background noise, which may affect the perception of alarm sounds (page 14; lines 501 ff.).
Comment 7: Industry participation can be further explained
Response 7: We thank the reviewer for this comment. We have clarified in the Methods (Ethical Considerations and Data Security) that all data custody and analyses were fully under academic control and conducted independently, with no involvement from Philips. Philips representatives reviewed the manuscript but had no access to raw data and did not influence study design, analysis, interpretation, or manuscript preparation. Yoko and Avery Senn were compensated for developing the alarm sounds, but not for the study itself. All other potential conflicts of interest of the academic authors are disclosed in the appropriate section. See also Response to Comment 6 of Reviewer 1.
Comment 8: Studies show that in various departments equipment may not be used if no training is provided, are you planning to develop trainings? in situ? online?
Response 8: We thank the reviewer for this comment. The refined alarm sounds were implemented via an update on Philips patient monitors, and users cannot choose between the conventional and refined sounds. As this change only affects the auditory environment without altering clinical workflows, no formal training, either in situ or online, was planned or deemed necessary.
Comment 9: Provide contextualization of use, define settings.
Response 9: The refined alarm sounds were implemented on Philips patient monitors in the operating room setting. The intervention affects the auditory environment perceived by anesthesia providers, who were the only participants in our study and primarily interact with the patient monitors in the OR; no conclusions can be drawn regarding surgeons or other staff. The intervention does not alter workflow or monitor functionality. Users cannot select between conventional and refined sounds; all alarm sounds were updated simultaneously as part of a standard Philips monitor software update.
Reviewer 3 Report
Comments and Suggestions for Authors
The study addresses a critical dimension of human-machine interaction in clinical safety alarm sound ergonomics. However the novelty is limited. It refines existing alarm sounds rather than proposing a novel paradigm (e.g., auditory icons or speech alarms).
The collaboration with some brands raises potential bias concerns. Author independence should be clarified further.
The study adds real-life validation of redesigned alarms post–clinical implementation, provides quantitative mixed-effects modeling of auditory appeal and recognition
It confirms rather than challenges prior hypotheses on the pleasantness–functionality trade-off.
However it does not significantly deepen the mechanistic understanding of alarm perception.
The pre–post design after real clinical implementation enhances translational value.
Single-center pre–post design is appropriate but lacks a control group, limiting causal inference (L373–L377).
The 4-month familiarization period is reasonable (L162) but could have been tested longitudinally.
Recruitment via convenience sampling may introduce selection bias (L132).
Dropout rate 14.4% is acceptable, though missing data handling is not described.
Use of Likert scales is justified but lacks validation (no Cronbach’s α or reliability).
Recognition were conducted via MacBook Air playback (L146–L149), which differ from clinical monitors, which could be a validity issue. Include audio calibration protocol to ensure acoustic fidelity between MacBook and clinical devices.
Non-inferiority margins (L220) lack empirical reference. Add justification, power analysis.
No correction for multiple testing is mentioned.
Address potential expectancy bias, as participants were aware of the intervention.
Results are well-organized and logically presented.
Conclusions are consistent with data.
Implications could be broadened about future standard updates or cross-platform alarm design.
Explore training implications for clinical staff adapting to new alarm systems.
References are generally appropriate and up-to-date
Figures 2 / 3 are clear, though axis labels, CI bars… could be added.
Add statistical annotation (p-value, CI line…) directly in figures.
Writing is fluent.
L108: “huma user-centered” = “human user-centered.”
Author Response
Dear Reviewer 3,
Point by Point Response
We would like to sincerely thank the three reviewers and the editor for their time and the intellectual effort they dedicated to our paper. We have addressed each comment below with great care and believe that the manuscript has become substantially more transparent and engaging for the reader as a result.
Reviewer 3
Comment 1: The study addresses a critical dimension of human-machine interaction in clinical safety alarm sound ergonomics. However the novelty is limited. It refines existing alarm sounds rather than proposing a novel paradigm (e.g., auditory icons or speech alarms).
Response 1: We thank the reviewer for this comment. We acknowledge that the refined alarm sounds do not represent a fundamentally novel paradigm, such as auditory icons or speech-based alarms, and we have already discussed this limitation in the Discussion section. Nevertheless, this study represents the first clinical evaluation of newly refined alarm sounds in a real operating room setting. Importantly, we observed the trade-off between improved perceived sound appeal and slightly reduced recognizability of low-priority versus medium-priority alarms. In practice, this may be of limited clinical significance, as low-priority alarms do not typically require immediate action from anesthesia providers and are not essential for direct clinical decision-making.
Comment 2: The collaboration with some brands raises potential bias concerns. Author independence should be clarified further.
Response 2: Please review the responses to Reviewer 1, Comment 6, and Reviewer 2, Comment 7.
Comment 3: The study adds real-life validation of redesigned alarms post–clinical implementation, provides quantitative mixed-effects modeling of auditory appeal and recognition. It confirms rather than challenges prior hypotheses on the pleasantness–functionality trade-off. However it does not significantly deepen the mechanistic understanding of alarm perception.
Response 3: We thank the reviewer for this insightful comment. We agree that our study was not designed to investigate the underlying mechanisms of alarm perception. Rather, it is a clinical evaluation comparing two sets of alarm sounds in real-life operating room conditions. One interesting observation concerning alarm perception is that reducing the harshness of low- and medium-priority alarms improves the overall auditory environment while maintaining recognizability of high-priority alarms. This is particularly relevant because high-priority alarms typically require immediate action from anesthesia providers, whereas lower-priority alarms often do not correspond to direct clinical tasks. Thus, our study provides practical insights into optimizing alarm design for clinical relevance, even if it does not directly address mechanistic questions.
Comment 4: The pre–post design after real clinical implementation enhances translational value. Single-center pre–post design is appropriate but lacks a control group, limiting causal inference (L373–L377).
Response 4: We thank the reviewer for this comment. We agree that conducting a pre–post study following real clinical implementation enhances the translational value of our findings, as they reflect actual operating room conditions. We also acknowledge that the single-center pre–post design, while appropriate for this initial evaluation, lacks a control group, which limits causal inference. Therefore, while we can report observed changes in alarm perception, we cannot definitively attribute these changes solely to the intervention. This limitation has been added to the manuscript in the Limitations section (see page 13, line 473 ff.).
Comment 5: The 4-month familiarization period is reasonable (L162) but could have been tested longitudinally.
Response 5: We thank the reviewer for this comment. We agree that a longitudinal evaluation of the familiarization period could provide additional insights. In our study, we implemented a 4-month familiarization period prior to data collection to ensure that participants were sufficiently exposed to the refined alarm sounds. While we did not formally assess changes over time during this period, doing so would have required substantially greater personnel and time resources. The pre–post design after the familiarization period allowed us to capture the effect of the intervention under stable, real-world conditions. This is a known limitation of pre–post study designs and is addressed in our limitations section.
Comment 6: Recruitment via convenience sampling may introduce selection bias (L132).
Response 6: We thank the reviewer for this comment. We have addressed this point in the Limitations section of the manuscript (see page 14, line 460 to 462).
Comment 7: Dropout rate 14.4% is acceptable, though missing data handling is not described.
Response 7: We thank the reviewer for this comment. Only participants who completed both the pre- and post-phase surveys were included in the analysis. There were no missing data within the surveys themselves, as all questions were mandatory and had to be answered before the survey could be submitted.
Comment 8: Use of Likert scales is justified but lacks validation (no Cronbach’s α or reliability).
Response 8: You are absolutely right. We have addressed this; please see our response to Reviewer 1, Comment 5.
Comment 9: Recognition were conducted via MacBook Air playback (L146–L149), which differ from clinical monitors, which could be a validity issue. Include audio calibration protocol to ensure acoustic fidelity between MacBook and clinical devices.
Response 9: You are also absolutely correct. We have incorporated this into the manuscript; please see our responses to Reviewer 1, Comment 3, and Reviewer 2, Comment 3.
Comment 10: Non-inferiority margins (L220) lack empirical reference. Add justification, power analysis.
Response 10: We have decided not to perform a non-inferiority analysis and instead report the confidence intervals for the outcomes. Please see our response to Reviewer 1, Comment 4.
Comment 11: No correction for multiple testing is mentioned.
Response 11: We thank the reviewer for this comment. As described in the Methods, only one predefined primary model (Perceived Sound Appeal) was tested for statistical significance. Therefore, correction for multiple testing was not applied. The additional models for Perceived Functionality and Recognizability were reported descriptively with corresponding confidence intervals, but without inferential testing. A correction for multiple comparisons would only be required if several hypotheses of equal importance were tested simultaneously, which was not the case in this study.
Comment 12: Address potential expectancy bias, as participants were aware of the intervention.
Response 12: We thank the reviewer for this comment. To address potential expectancy bias, we conducted a sensitivity re-analysis including only participants who did not attend the training session. No differences were observed compared to the full sample. The results of this re-analysis are provided in Multimedia Supplement 3. Please also see our response to Reviewer 1, Comment 1.
Comment 13: Results are well-organized and logically presented.Conclusions are consistent with data. Implications could be broadened about future standard updates or cross-platform alarm design.
Response 13: We thank the reviewer for this positive comment. We have added a brief discussion on the potential implications for future standard updates and cross-platform alarm design (see page 14, line 4459 ff.).
Comment 14: Explore training implications for clinical staff adapting to new alarm systems.
Response 14: We thank the reviewer for this comment. Initially, we considered that formal training was not necessary, as the changes to the alarm sounds were minor and, in practice, standard updates of patient monitors typically do not involve training. Moreover, our sensitivity re-analysis including only participants who did not attend the training session showed no differences in outcomes, further suggesting that additional training would unlikely have led to different adaptation by the participants. We nevertheless fully agree with the reviewer that for more complex system changes, dedicated user training would be essential, as otherwise the adoption rate would likely remain very low.
Comment 15: References are generally appropriate and up-to-date.
Response 15: Thank you
Comment 16: Figures 2 / 3 are clear, though axis labels, CI bars… could be added. Add statistical annotation (p-value, CI line…) directly in figures.
Response 16: We thank the reviewer for this comment. Figures 2 and 3 present descriptive statistics only. The results of the linear mixed regression models for Perceived Sound Appeal and Perceived Functionality, as well as the mixed Poisson regression model for Recognizability, are reported in the text with corresponding p-values and confidence intervals. We chose not to visualize the model results in the figures to avoid double reporting.
Comment 17: Writing is fluent.
Response 17: Thank you.
Comment 18: L108: “huma user-centered” = “human user-centered.”
Response 18: Thank you very much. We have corrected the spelling error.
Round 2
Reviewer 1 Report
Comments and Suggestions for Authors
Thank you for addressing the comments